# Advances in Ultrasonic Spray Pyrolysis Processing of Noble Metal Nanoparticles—Review

**DOI:** 10.3390/ma13163485

**Published:** 2020-08-07

**Authors:** Peter Majerič, Rebeka Rudolf

**Affiliations:** 1Faculty of Mechanical Engineering, University of Maribor, Smetanova Ulica 17, 2000 Maribor, Slovenia; peter.majeric@um.si; 2Zlatarna Celje d.o.o., Kersnikova 19, 3000 Celje, Slovenia

**Keywords:** Ultrasonic Spray Pyrolysis, Nanotechnology, noble metals nanoparticles, formation mechanism, characterisation, properties

## Abstract

In the field of synthesis and processing of noble metal nanoparticles, the study of the bottom-up method, called Ultrasonic Spray Pyrolysis (USP), is becoming increasingly important. This review analyses briefly the features of USP, to underline the physical, chemical and technological characteristics for producing nanoparticles and nanoparticle composites with Au and Ag. The main aim is to understand USP parameters, which are responsible for nanoparticle formation. There are two nanoparticle formation mechanisms in USP: Droplet-To-Particle (DTP) and Gas-To-Particle (GTP). This review shows how the USP process is able to produce Au, Ag/TiO_2_, Au/TiO_2_, Au/Fe_2_O_3_ and Ag/(Y_0.95_ Eu_0.05_)_2_O_3_ nanoparticles, and presents the mechanisms of formation for a particular type of nanoparticle. Namely, the presented Au and Ag nanoparticles are intended for use in nanomedicine, sensing applications, electrochemical devices and catalysis, in order to benefit from their properties, which cannot be achieved with identical bulk materials. The development of new noble metal nanoparticles with USP is a constant goal in Nanotechnology, with the objective to obtain increasingly predictable final properties of nanoparticles.

## 1. Introduction

The processing of aerosols (solid or liquid particles suspended in gas) and obtaining various types of fine particles and nanomaterials from aerosols has been known for a long time. These materials are meant for diverse uses in different fields, from nanomedicine, sensing applications, electrochemical devices, catalysis, energy conversion and storage, to name a few [1,2,3,4]. Generally, the basic principle of aerosol formation arises from a solution that contains dissolved material (usually salts of various metals)—which are commonly called precursors. In the synthesis, the powder product is atomised from an aerosol composed of liquid droplets, which are then transported with a carrier gas to the reactor where the chemical transformation occurs. The chemical composition of the precursor depends on the chemical composition of the fine powder we want to obtain and the type of processing method used, such as spray drying, Spray Pyrolysis, Flame Spray Pyrolysis, thermal decomposition, micronisation or gas atomisation [5,6,7].

Spray drying techniques are used widely in the food industry, such as for the production of milk or fruit powders and for flavour ingredients, as well as in pharmaceutics’ production [5,8]. This technique is a low-cost, environmentally friendly method with proven scalability, and is well known in Chemical Engineering [9].

Spray Pyrolysis, Flame Spray Pyrolysis and aerosol decomposition or thermolysis, generally involve atomisation of metal salt solutions as the precursor for the final metal or metallic oxide particles, and subjecting these liquid droplets to a high temperature in a furnace or a flame. Water or organic solutions with different metal salts may be used, from chlorides, acetates, nitrates, bromides, sulphides, hydroxides, etc., depending on the necessary composition and structure of the final particles [5,10].

Aerosol assisted sol–gel processes combine the sol–gel chemistry with aerosol processing [11]. In principle, each aqueous aerosol droplet may be used as a single, micron-sized sol–gel reactor, capable of producing different particle structures, such as mesoporous particles of various materials and chemical compounds [12,13,14]. In gas atomisation, liquid metal is sprayed through a nozzle and cooled rapidly to produce solid small spherical particles of the used metal or alloy [6,7].

Most of these techniques require the atomisation of a liquid solution into micron-sized droplets. The generation of such droplets is achievable by using pressure, electrostatic fields, centrifugal force or ultrasound [5].

This review is focused on Ultrasonic Spray Pyrolysis (USP), which combines the ultrasound used for dispersing the precursor solution into droplets and chemical decomposition of the dissolved material inside the droplets at elevated temperatures, resulting in the formation of fine powder. In the USP reactor the evaporation of solvent from droplets occurs first, in the next stage the remaining solvent inside the dry droplets is decomposed chemically via pyrolysis, and, in the final stage, fine particles are obtained through the sintering process. The advantage of the USP method is the simplicity of setting up individual process segments and changing their configuration, continuous nanomaterials’ synthesis, and the possibility of synthesising nanoparticles from various raw materials. The disadvantage is the low efficiency, due to losses of the dissolved material on the construction elements of the USP device.

Currently, there is a need for scaled-up production of nanoparticles, able to take up the increasing quantities of nanoparticles required for advanced applications. Technologies for generating nanoparticle powders and suspensions have existed for several decades [15], and are being improved upon continually, while novel approaches are also being studied [16,17]. One potentially reliable method for the large scale synthesis of nanoparticles is the USP process [18]. This process is considered to be relatively cost-effective, and easily scalable from the laboratory to an industrial level. Several different types of nanoparticles and structures can also be produced by USP, such as solid or hollow nanoparticles, core–shell and ball-in-ball structures, etc. [4,19,20,21,22], giving an additional advantage to this production method. Yolk–shell-structured [23,24], mesoporous [25] and multicomponent composite particles [26,27] were also proven to be synthesised with this method, by a group focusing on Li-ion and Na-ion battery applications of these particles. Depending on the particle composition, the formation of mesoporous or multicomponent structures can also be facilitated with subsequent processing of the initial particles produced by USP. The process is also used extensively for thin film deposition [21,28], with multiple layers [29]. The latest scientific finding reported [30] that a fully solution-based fabrication process exploiting USP and spin coating techniques, owing to their simplicity, high degree of freedom for mixing metal oxide precursor salts, and larger area deposition, opens the potential application of solution-processed transistors for low-cost electronic devices.

In our long-term investigation and experimental work, we have used USP to produce several metallic and metal oxide particles with micron and nano sizes, ranging from Au (from Au salts and from gold scrap), Ag, Ag/TiO_2_, Au/TiO_2_, ZnO, Fe/Au, Co, Cr and rare earth metals. For Au nanoparticles (AuNPs) we have also investigated extensively the cytotoxic and immunomodulatory properties of these particles for biomedical applications [31,32,33], as well as their behaviour when exposed to in situ heating during TEM observation [34], and their use in printing inks for Electronics (not yet published). We have also applied Au and ZnO nanoparticles in PMMA composites for advanced dental materials for removable prostheses [35]. The Ag/TiO_2_ and Au/TiO_2_ particles were produced for use as catalysts, and rare earth metals for photocatalytic applications [36,37].

In this review, we outline the advances and insights made on the USP process, obtained through theoretical and experimental investigations on the addressed materials, completed over the years. Furthermore, we search literature from the database—the last search was done on 12 May 2020. This overview contributes to a more comprehensive understanding of this process for similar endeavours on the USP technique, and demonstrates the synthesis and characterisation of some characteristic nanoparticles through USP.

## 2. Ultrasonic Spray Pyrolysis

The main elements of the conventional USP device are the ultrasonic generator or nebuliser, the reactor furnace, and a system for nanoparticle collection and cooling (Figure 1). Ultrasonic nebulisers are the most efficient amongst other types of nebulisers for nanomaterial production, such as pneumatic and electrostatic, while also being affordable and having a low droplet velocity. Pneumatic nebulisers produce droplets by expanding a pressurised liquid. These nebulisers have a high droplet output of large sizes around ≈50 µm, and the droplet sizes and size distribution are difficult to control [38]. Electrostatic nebulisers use an electrostatic field to extract droplets from a liquid [28]. They have a low productivity, and can only be used on conductive liquids. Ultrasonic nebulisers produce droplets in a narrow size distribution, less than 10 µm, with acceptable productivity. Ultrasonic spray nozzles generate more stable and uniform droplets, but cannot produce droplets as small as ultrasonic nebulisers [38]. As such, ultrasonic nebulisers have good characteristics regarding their droplet output, and are used commonly in Spray Pyrolysis processes (Table 1).

The sizes of the synthesised nanoparticles depend on the ultrasound frequency [4,41,42], which determines the sizes of the aerosol droplets and the concentration of the dissolved salts in the precursor solution droplets. Due to vibrations of the ultrasound below the solution surface, the kinetic energy of the solution molecules increases rapidly. This causes small droplets to overcome the surface tension and break away from it. This effect, known as nebulisation (Figure 1), produces micron-sized aerosol droplets, which act as individual chemical reactors when subjected to thermal treatment [43,44]. Droplets in a size distribution from 1 to 15 µm are created with a high-frequency ultrasound (0.5–3 MHz) [45].

The generated droplets of the precursor solution are transported into the furnace with a carrier gas, where the synthesis stages of evaporation and droplet shrinkage, thermal decomposition and densification take place. Depending on the precursor salt composition and chemistry, a reaction gas may be included with the carrier gas, in order to promote the formation of pure metal or metal oxide nanoparticles. The process forms solid, nonporous nanoparticles with different amounts of aggregation, depending on the process conditions and parameters used (reaction temperature, gas flow, residence time, precursor selection and concentration).

One classification for nanoparticle synthesis processes is based on the physical state from which the nanoparticles are formed: From the gas, liquid or solid state. Depending on the type of precursor and material synthesised, USP can be categorised in all three states, as nanoparticles can be formed from the gas or the liquid/solid state. Depending on the physical state from which nanoparticles are formed with USP, there are two well-known main conversion routes described in the literature [4,18,48]: the Droplet-To-Particle (DTP) and Gas-To-Particle (GTP) conversion mechanisms. These formation mechanisms can both occur during synthesis with USP, and are determined by the various USP parameters—aerosol droplet size, gas flow, reaction temperatures, precursor solution salt and solvent volatility (ease of vapourisation), etc.

### USP Parameter Selection

The size of the obtained nanoparticles is related to the droplet diameter and the initial concentration of the precursor solution. Increasing the ultrasonic frequency reduces the droplet diameter of the produced aerosol, and also increases the ratio of smaller nanoparticles in the final product. The formation of droplets by ultrasound was first described by Wood and Loomis in 1927 [49]. In 1962, Lang formulated an equation to describe the connection between the ultrasonic frequency and the mean droplet diameter [50]. A theoretical equation for the prediction of sizes of obtained nanoparticles was formed in previous work [51], combining Kelvin’s equation and an equation based on the capillary theory.
(1)$$D_{mat}=0.34\cdot \sqrt[{3}]{\left(\frac{8\cdot \pi \cdot \gamma \cdot C_{sol}\cdot M_{mat}}{\rho_{sol}\cdot f^{2}\cdot \rho_{mat}\cdot M_{sol}}\right)}$$
$D_{mat}$ is the final particle diameter; $C_{sol}$, $\rho_{sol}$ and $M_{sol}$ the solute concentration, density and molar mass; $M_{mat}$ and $\rho_{mat}$ are the molar mass and density of the desired material.

In this equation, the size of the nanoparticle depends mainly on the diameter of the droplet (frequency of the ultrasonic atomiser) and on the desired material concentration of the precursor solution. Unfortunately, the coalescence of the droplets is difficult to prevent, and the calculated mean droplet diameter was expected to be somewhat smaller than the actual size of the droplets in the performed experiments. In the mentioned equation, there is also no account for the effect of temperature on nanoparticle size. An example of calculating nanoparticle size is given for the production of AuNPs with an ultrasound frequency of 0.8 and 2.5 MHz in Figure 2.

A concentration of Au between 1 and 20 g/L was selected for this calculation. Hence, for an ultrasonic frequency of 0.8 MHz, the calculated mean AuNP diameter ranged from 150 to 407 nm. Increasing the frequency up to *f* = 2.5 MHz, the calculated mean particle diameter was between 70 and 190 nm. The increase in frequency resulted in smaller droplet diameter and smaller nanoparticle size.

In practical terms, the aerosol droplets are actually generated in a size distribution, rather than in a single size; however, the theoretical calculations provide a good starting point for the selection of the precursor concentration. For a more exact determination of the aerosol droplets, laser diffraction measurements are needed of the produced aerosol mist [45]. As an example, theoretical calculations predicted a size of 193 nm for gold nanoparticles with a 0.8 MHz ultrasound, and 91 nm with a 2.5 MHz ultrasound. The experimental results showed a size range from 38–200 nm for 0.8 MHz and 10–250 nm for 2.5 MHz [46]. The aerosol collisions, droplet size distribution and gas-phase formation are not included in the theoretical calculation. However, the calculations are still adequate to be used as a guideline for the selection of precursor concentration.

The selected reaction temperatures depend on the decomposition temperatures of the precursor components. In relation to gas flow, the temperature profile along the length of the reaction tube changes—this needs to be considered while designing the production of nanoparticles, so that the residence time of the particles allows for the total decomposition to be carried out, with no unreacted particles in the final product. As such, the residence time, and the velocity of the aerosol inside the reaction tube, also depend on the tube diameter. The gas flow, tube diameter and length also dictate the velocity profile and temperature profile across the cross-section of the tube, as is known from fluid dynamics. Tubes of larger diameters and shorter lengths have an undeveloped gas flow, while longer tubes with smaller diameters have a more developed flow, as shown in Figure 3. The short tube lengths from Figure 3 are usually not used with USP, but the velocity profile aspect is important to consider when selecting the gas flow, residence time and temperature parameters, which dictate whether the aerosol will transform fully into the desired particles.

Adding a reaction gas into the tube also adds additional occurrences of localised turbulent swirls of the travelling gas, and, as a result, there is more uncertainty regarding the formation mechanisms of the particles in the tube. A fully developed gas flow inside the reaction tube yields more uniform particles. The temperature profile along the length and cross-section of the tube is related closely to the gas flow conditions. More stable flow currents again allow for a more controlled temperature inside the tube and less inconsistency in the particle formation.

Temperature, tube dimensions, gas flow, precursor concentration and ultrasound frequency present a complex system of interdependent USP aspects that affect nanoparticle formation. These parameters leave plenty of manoeuvering room for improvements and optimisation for a given particle formation inside the USP, although one must consider what changes are introduced when one parameter is altered, and what effect this alteration has on the nanoparticle formation conditions.

An important element to consider is also the selection of the precursor salt and its effect on the particle formation. Most frequently, metal-containing inorganic compounds are used, such as chlorides, nitrates and sulfates [38,52], as they have a lower volatility and are more prone to form particles from the Droplet-To-Particle mechanism, discussed in the next chapter. Metal-organic and organometallic compounds such as acetates, oxalates, carbonates and alkoxides are more volatile, promoting the Gas-To-Particle conversion mechanism in USP [38,52]. The choice of solvent likewise facilitates the presence of liquid or gas phases in the system. The most common solvent is water, while alcohol, ionic liquids and organics are used in order to modify the volatility in favour of the gas particle conversion [38].

Some particles have the tendency to agglomerate in the collection system after synthesis, due to their high surface energies. For these nanoparticles (in general for all such nanomaterials), it is also imperative to prevent their agglomeration with the use of suitable collection mediums and stabilisers.

## 3. Particle Formation with USP

### 3.1. Droplet-To-Particle Conversion Mechanism in USP

The liquid-to-solid and solid-to-solid conversion processes with USP can be described with the DTP mechanism. This mechanism, in general, consists of the formation of droplets, transportation of these droplets into a heating zone, evaporation of the solvent and thermal conversion of the solute into the final nanoparticles. As the droplet with dissolved material is being evaporated it shrinks, and, simultaneously, increases the mass fraction of the solute inside the droplet. The solute can begin to precipitate before uniform saturation is reached across the droplet because the solute diffusion is slower than the evaporation of the solvent. As the solid material is being precipitated on the droplet surface due to supersaturation, the liquid can become trapped in the centre. It then begins to evaporate through the newly formed surface crust (Figure 4). This slows down the evaporation rate.

Solute precipitation in droplets has not been described by any theory in a quantitative manner. For a given solute, the supersaturation required for precipitation must be measured [4,54], and is a function of the exact composition of the solution (impurities act as precipitation sites). Because the rate of nucleation determines particle morphology, the evaporation determines particle morphology (evaporation depends on a number of factors, such as surrounding vapour pressure and temperature).

Intraparticle reactions, such as thermal decomposition, also occur in the aerosol before and after solvent evaporation, and can influence particle morphology. These were studied with thermogravimetric analysis, differential thermal analysis and differential scanning calorimetry [4]. The precursor characteristics determine whether hollow or porous particles are formed. Whether a precursor melts or not before reacting has a very distinct difference on the final nanoparticle morphology. The volume fraction of the solute also influences the formation of porous or hollow particles, as does a high reduction of volume because of reactions, shown in Figure 5. An example of the particles formed by the DTP mechanism is shown in Figure 7.

### 3.2. Gas-To-Particle Conversion Mechanism in USP

The GTP formation mechanism generally follows the supersaturation of a gaseous species of the desired material, which causes nucleation and new particle formation. The particles can be created by chemical reactions of gaseous precursors, or by physical processes, such as cooling of hot vapour. The average particle diameter, total particle concentration, size distribution and particle morphology evolve along the aerosol reactor in two different modes. One mode is nucleation–condensation, where a monomer (a gas-phase molecule) is formed by a chemical reaction or a decrease in temperature until nucleation occurs. The saturation ratio increases, and growth of the monomer proceeds by condensation of the monomer onto the particles. When no collisions occur, they can remain nearly spherical through the aerosol reactor path. Surface reaction on the particles may also occur, promoting growth. Another mode is nucleation–coagulation, where particles are formed and their high concentration allows collisions and coalescence of particles, resulting in the growth of the particles. Both modes can be found in laboratory processes, while nucleation–condensation is usually not found in industrial systems.

In the nucleation–condensation mode, the morphology of the particles depends on collisions and coalescence. Particles coalesce by sintering after collisions in order to become spherical. The ratio of rates of collisions and coalescence (α_c_ [4,56,57]) determines the morphology. In collision-limited growth (α_c_ = ∞), the sintering rate is rapid relative to collisions, allowing for the formation of spherical particles between collisions. In sintering-limited growth (α_c_ = 0) the particles exist as aggregates. Intermediate conditions (α_c_ ≈ 1) are in existence in real situations, where the particle morphology is a function of parameters that control the sintering rate—temperature and material properties. The primary particle size is also a function of these parameters (Figure 6).

When the aerosol droplet evaporates within the GTP mechanism, the solute is vapourised along with the solvent, resulting in the presence of the solute vapours and their partial vapour pressure. When saturation of vapours is reached, nucleation occurs, and growth of particles proceeds as described previously. Particles formed by the GTP conversion are typically smaller than the particles formed by the DTP mechanism. A comparison of the particles formed by these two mechanisms is shown in Figure 8.

## 4. Synthesis of AuNPs

### 4.1. Gold Chloride Precursor

There are various raw materials which can be used for preparing precursor solutions for AuNPs’ synthesis (compounds containing Au: AuBr_3_, HAu(NO_3_)_4_, Au(O_2_CCH_3_)_3_). In our case, gold chloride or tetrachloroauric acid HAuCl_4_(s) was investigated the most extensively with USP synthesis, due to its price, availability and chemical stability. The precursor solution was prepared by dissolving HAuCl_4_ in water [31,46]. It was also confirmed that AuNPs’ synthesis is possible from homemade H-HAuCl_4_, completed through the chlorine gas method by using HCl and KMnO_4_ for dissolving a gold pellet in water [58].

By using concentrations of dissolved Au from 0.5 g/L to 5.0 g/L Au in the precursor solution, each droplet contains such an amount of material that, after evaporation and drying, nanoparticles are formed with diameters from a few 10 nm (at 0.5 g/L Au) up to more than 100 nm (at 5.0 g/L Au in the precursor solution).

Droplets of the starting solution are transported into the furnace with a carrier gas. Inside the furnace, the AuNPs are formed according to the following synthesis stages:Evaporation and droplet shrinkage (HAuCl_4_ with water)—above 100 °C;Thermal decomposition of HAuCl_4_ into AuCl_3_—at 258 °C (theoretical);Reduction of AuCl_3_ with hydrogen gas and the formation of Au—above 300 °C;Densification (sintering processes).

The listed synthesis stages are taking place in a disordered fashion inside the USP tube furnace, as different phases of the starting material are present simultaneously at various points in the process. With the initially smaller diameters of the aerosol droplets with diameters ≈1 μm, nanoparticles are formed much sooner than with larger droplets with diameters ≤10 μm. Different droplet sizes are inherent in the aerosol droplet generation, as the nebuliser produces droplets in a size distribution, rather than uniformly sized droplets. The droplets are also ejected from the solution in a mist, making collisions between droplets difficult to prevent during gas transportation.

Therefore, some solid nanoparticles have already formed in the reactor tube, while some aerosol droplets have not yet evaporated at the same point inside the reactor tube. This results in synthesising various sizes and regular and irregular shapes of nanoparticles, due to aerosol collisions and coagulation. This is not suitable for synthesising more uniform AuNPs.

With the conventional USP, we have synthesised AuNPs with sizes ranging from 10 to 300 nm, with different shapes, from spherical, irregular, triangular and cylindrical in a single batch [46,47]. Such AuNPs were not suitable for fine-tuning of their properties (Figure 7).

In order to alleviate this phenomenon and produce more uniform AuNPs in size and shape, the reaction gas was introduced at a later stage in the tube reactor (modified USP). This allows for droplet evaporation and aerosol particle drying before thermal decomposition and reactions take place for final particle formation. In this way, more control could be achieved over the rate of droplet evaporation and the rate of precursor salt diffusion inside the droplet. These rates with USP synthesis depend on several factors: precursor solution concentration, droplet sizes, number of droplets and relative humidity in the system, velocity of droplet transportation inside the furnace, pressure in the system, dimensions of the transport pipes and temperature profile inside the furnace. For setting up suitable parameters (precursor solution concentration, gas flow, furnace temperature), information is needed for the starting solution properties, such as density and surface tension and characteristics of the dissolved HAuCl_4_ diffusion inside the solution and AuNP growth.

Several experiments with different parameters were performed with the modified USP. The synthesised AuNPs were characterised with various characterisation techniques for identification of their sizes, shapes, chemical composition and degree of agglomeration. Based on these results, we have surmised the influence of individual USP parameters on the AuNP formation mechanisms. We have identified that the AuNPs are formed from droplets and from the gas phase. This means that they are formed from a combination of DTP and GTP formation mechanisms (Figure 8) [47,59]. Gold chloride is highly hygroscopic, acidic and a highly volatile gold salt. Using this salt in the USP process requires great control over the process, in order to produce AuNPs of the desired shapes and sizes, due to the mixture of gas and solid phases present in the reaction tube, as well as the inherent physical and chemical phenomena that are being performed inside the tube furnace (collisions, coalescence, aggregation, agglomeration, etc.).

In order to obtain uniform nanoparticles with such highly volatile precursors, a balancing act is needed between the physical events of the aerosol, governed by the USP parameters, and the chemistry of the used formulations. A helpful representation of the effect of these parameters on the particle growth and morphology is the residence time of the aerosol particles in the reaction furnace, as shown in Figure 9. Initially, primary particles are formed from the precursor. These particles may then be joined closely to each other by chemical forces (covalent, ionic), forming necks and hard agglomerates or aggregates, which are difficult to break apart into primary particles. Soft agglomerates (simply called agglomerates) are composed of loosely attached primary particles or aggregates by physical forces (electrostatic, van der Waals), and can be dispersed into individual elements by applying force or energy (e.g., ultrasound) [10]. The length of each particle growth stage or how long the residence time of the particles is in each stage of coalescence, aggregation, or agglomeration, determines the final particle morphology.

Our experimental work on gold chloride showed agreement with aerosol processing morphology models [10]. Having a high residence time produced a mixture of primary small particles, with larger aggregates, along with larger particles and agglomerates in a single batch. Decreasing the residence time led to more uniformly sized, spherical AuNPs. The main condition for producing uniform, unaggregated particles is the length of the aggregation stage. Depending on the length of the aggregation stage, two growth routes are shown in Figure 9. A longer residence time in this stage enables sintering at elevated temperatures and the formation of irregularly shaped aggregates. Reducing the length of this stage—i.e., cooling the primary particles shortly after their formation, produces more fine, spherical AuNPs.

A practical model was drawn (Figure 10), showing the dependence of two influential USP parameters, gold concentration in the precursor solution and gas flow. The gold concentration in the precursor solution constrains the gold salt solid and gas phase content in the process. The gas flow determines the residence time, the droplet evaporation and particle reactions’ times. Synthesis of targeted spherical AuNPs with a size distribution around 50 nm was achieved with these findings [59].

### 4.2. Gold Acetate Precursor

Solid phase metal acetate precursors, such as Au (III) acetate, Au(O_2_CCH_3_)_3_), have shown great potential to expand the composition and structures of nanoparticles. Gold acetate was always readily available, even though its low solubility in most solvents limits its use in the synthesis of AuNPs [17,60].

The chemistry of this salt and its effects on synthesis mechanisms in the USP were investigated with several experimental trials for producing AuNPs. The main byproduct of gold acetate decomposition is acetic acid [61], which further influences the morphology of the formed AuNPs, showing clear differences when compared to the AuNP structures formed with gold chloride. Gold acetate was selected as an alternative precursor to gold chloride for USP synthesis of AuNPs, with the aim of obtaining spherical AuNPs with a narrower size distribution and higher concentration.

The results showed a much smaller particle size when compared to gold chloride AuNPs, along with the formation of mesh-like agglomeration structures (Figure 11). This suggests that the addition of acetic acid in the aerosol flow induces the formation mechanisms to follow a route of primary particle formation and soft or hard agglomeration of these nanoparticles, depending on the synthesis conditions.

### 4.3. Gold Nitrate Precursor

Gold nitrate, HAu(NO_3_)_4_, was chosen as an additional precursor for AuNP synthesis with USP, and for comparison of the produced nanoparticle morphology. Commercially available gold nitrate is poorly soluble in water, and an investigation was needed for preparing a soluble and chemically stable water-based gold nitrate precursor [34]. The preparation of a precursor solution required reflux boiling and additional steps to make it usable within the USP process.

The results showed highly spherical AuNPs with a mean size of about 174 nm at a precursor starting concentration of 2.5 g of dissolved Au per litre. The nanoparticles were also hollow, which is a structure not obtained with the previously used gold salts.

With the different precursor salts used for AuNP synthesis with USP up to date, a general overview of the morphologies of the nanoparticles can be given, as shown in Figure 12.

## 5. Synthesis of Particles Decorated with Nanoparticles

Nanoparticles of different materials have varied properties that make them interesting for implementation in novel products. Metallic nanoparticles have enhanced optical properties due to Surface Plasmon Resonance (Ag, Au [1]), while they may also have antimicrobial (ZnO [62]), magnetic (Fe_2_O_3_ [1]), catalytic properties (Ag [63]), or a high electrochemical capacity for electrode materials (LiFePO_4_-C [64]), able to be used in biotechnology, magnetic separation processes, sensors or electronic and energy devices. By combining two different nanomaterials, it is possible to combine and improve their characteristics and increase their potential application. One of the most common ways to achieve this is by using binary systems.

Bimetallic particles are functional materials for applications, produced by joining two elements in some type of construction, such as core–shell, alloy or otherwise [65,66]. The combined features of the two materials can result in new effects and properties, or improve existing ones, such as enhanced catalysis or tunable plasmonic properties. These composite structures may be used for various novel applications in catalytic processes [65], sensor devices [65], Electronics, for Magnetic Resonance Imaging [67,68,69], photothermal treatment of cancer [70] and drug delivery systems [67,68,69].

The flexibility of the USP process allows it to be used for the production of such composite particles by dissolving several precursor salts in the starting solution. It is also possible to use several ultrasonic generators, and introduce the produced aerosol into the same material flow. As such, we have used USP to produce complex Ag/TiO_2_, Au/TiO_2_ and Au/Fe_2_O_3_ nanoparticles.

### 5.1. Ag/TiO_2_ and Au/TiO_2_ Nanoparticles

The produced metal/oxide Ag/TiO_2_ and Au/TiO_2_ nanoparticles were produced by USP for use in photocatalytic applications. The catalytic properties of the Ag and Au nanoparticles are enhanced by the oxide substrate. The demands for optimal catalytic properties are small noble metal nanoparticle size and low content of noble metal in the shell (up to a maximum of 10%) [38]. TiO_2_ was selected as the Ag and AuNP substrate or core material due to its inertness, chemical stability, nontoxicity [71] and antimicrobial properties [72,73].

The selected precursor for TiO_2_ particle formation was tetra-*n*-butylorthotitanate, C_16_H_36_O_4_Ti, while gold chloride, HAuCl_4_ and silver nitrate, AgNO_3,_ were used for synthesis of the Au and Ag particles. The titanium-containing compound C_16_H_36_O_4_Ti was first stabilised with hydrochloric acid or nitric acid for the precursor solution preparation, as it hydrolyses and decomposes in contact with water. Gold chloride or silver nitrate were added into the prepared titanium precursor solution in various concentration ratios. The final prepared solution was then used in the USP process for Ag/TiO_2_ and Au/TiO_2_ nanoparticle synthesis at temperatures up to 1000 °C [36].

Figure 13 shows examples of the Ag/TiO_2_ nanoparticles obtained with USP at 800 and 1000 °C. Irregular morphologies of the oxide particles are visible, with the oxide particles being covered with Ag nanoparticles. The distribution of the Ag nanoparticles on the surface areas of the oxide is not homogeneous when a lower Ag concentration is used in the precursor solution. When a higher Ag concentration is used in the precursor solution, the coverage of Ag nanoparticles on top of oxide particles is more uniform. This was also the case when higher synthesis temperatures were used (1000 °C). The increase in temperature is believed to have a positive influence on the particle formation kinetics and precursor salt diffusion inside the aerosol droplet, since it facilitates the Ag nanoparticle distribution on the oxide surface. Higher Ag concentrations and higher temperatures also cause the Ag nanoparticles to be formed in more irregular shapes as compared to other experiments, where the spherical shape of these nanoparticles was present.

The produced TiO_2_ particles had a spherical solid shape with sizes ranging from 200 to 300 nm. The Ag nanoparticles were mostly spherical with an average size of around 50 nm [36,37].

Figure 14 shows examples of the Au/TiO_2_ nanoparticles obtained with USP at 550, 800 and 1000 °C. The core TiO_2_ particle sizes were similar in all of the experiments, while the AuNPs formed in irregular, spherical cubic and triangular shapes, in sizes around 10 nm. The main distinction is the AuNP coverage on the core TiO_2_ particles.

Focus Ion Beam (FIB) milling showed the presence of AuNPs inside the TiO_2_ particles, while, for Ag, this was not the case. About 30% of AuNPs were distributed randomly inside the TiO_2_ particle volume. More than 90% of Ag particles were distributed on the surfaces of the TiO_2_ particles [37].

#### 5.1.1. Metal Nanoparticle Nucleation on Oxide Particles

In order to have control over the final microstructure, it is important to understand the formation mechanism of these complex nanoparticles, which is led by nucleation and growth [39]. TiO_2_ is a crystalline material with three polymorphic phases, namely anatase (tetragonal), rutile (tetragonal) and brookite (orthorhombic). Different polymorphs present different nucleation sites for noble metals [74,75,76].

It was found that nucleation of metal nanoclusters initiated at surface defects, which is thermodynamically favourable due to the high adsorption energy and Ti cation sites between bridging oxygen rows [75]. The most stable surface for rutile is suggested as (1 1 0) [74], while for anatase it is identified as (1 0 1) [76]. Defects and oxygen deficiencies in the TiO_2_ crystallinity also play an important role as nucleation sites for the growth of noble metal nanoparticles or clusters. The equilibrium shape of a cluster is determined by the balance of the forces between the surface, the interfacial energy and wetting behaviour of the cluster [37].

Different metals have different interactions with the TiO_2_ surface, resulting in various microstructures of the clusters, depending on the metal type. The surface and crystallographic properties of the TiO_2_ core particle in relationship with the noble metal nanoparticles and the interface between the two components are important considerations for the final core–shell microstructure [37].

#### 5.1.2. Metal/Oxide Nanoparticles’ Formation Model

The noble metal/oxide particle formation with USP follows the same synthesis stage principles as when using a single precursor [18,19]. As the two components in the aerosol droplets are different in composition, they begin to diverge in the time frames of when the USP process stages of evaporation, drying and decomposition occur. Assuming that the two components have minimal influence on the synthesis stages of each other, a new formation model was proposed for synthesising core–shell nanoparticles. The model in Figure 15 is based on solute diffusion to the droplet surface and achieving a critical supersaturation concentration for nucleation.

The change in saturation concentration and position of metals and oxides in the dynamically-changing droplet/particle volume was proposed as a function of reaction time [37]. The different reaction times in the model are labelled as t0, t′, t″, t‴, where t0 < t′ < t″ < t‴. The reaction times have corresponding droplet/particle radiuses R, R′, R″, R‴. Ccrit m is the critical supersaturating concentration of the metallic precursor; Cs m is the saturation concentration of the metallic precursor; Cs ox is the saturation concentration of the oxide precursor; C m, C′ m, C″ m and C‴ m are concentrations of the metallic precursor at their corresponding times and radiuses. C ox, C′ ox, C″ ox and C‴ ox are concentrations of the oxide precursor at their corresponding times and radiuses.

As the process advances, the droplet/particle radius decreases from R to R‴. The concentration profiles inside the aerosol droplet/particle are shown under the aerosol in Figure 15. Initially, we assumed a homogeneous concentration distribution of the two precursor components throughout the droplet. The droplet begins to evaporate at elevated temperatures at the given time t′, decreasing its diameter and increasing the concentration of the components at the droplet surface. The oxide precursor concentration reaches its critical supersaturation (Ccrit ox) earlier than the metal precursor, as it has a much higher initial concentration in the solution. The precipitation of the oxide component starts firstly on the droplet surface. Since the concentration of the metal precursor had not reached the critical supersaturated concentration yet, the metal component diffused with the solvent to the droplet surface without any precipitation.

As the synthesis progressed further to reaction time t″, the radius decreased further, the oxide precipitation moved through the centre of the droplet, and the metal precursor reached the critical supersaturating concentration (Ccrit m). The metal precursor is now precipitating on the droplet surface. As identified earlier, the precipitation and nanoparticle growth favours surfaces with lower effective surface energy, such as the grain boundary, defects, etc.

In the last time frame at point t‴, almost all of the solvent had evaporated, and the droplet/particle had reached its smallest radius R‴. In an ideal case, all metallic precursors had succeeded in diffusing with the solvent to the droplet/particle surface and precipitated there. In the case that the precipitation front was moving faster than the diffusion front, some of the metal precipitation may also occur in the particle volume. This may occur at the relatively low value of Ccrit m, or, in the case of very small droplets, due to solvent evaporation in the droplet/particle volume [37].

In the proposed model, the critical supersaturation concentration defines at which point the precipitation of the individual component is going to take place. This critical concentration depends upon the solubility and the molar ratio of the precursor component in the aerosol droplet. Changing these values gives options for influencing the final metal/oxide core–shell particle formation. As solubility is determined by the chemistry and composition of the individual precursor components, this leaves the molar ratio as the determining factor able to be fine-tuned with USP. To achieve the target metal/oxide morphology, the concentration of the metal precursor should be lower than the concentration of the oxide precursor in the starting solution. This causes the oxide to precipitate earlier than the metal and form a substrate for the metal particles. After this precipitation, the usual synthesis stages of USP take place (completion of evaporation, decomposition/reaction, densification). This arrangement proposes that the concentration ratios of the two components govern whether a core–shell microstructure is formed, or whether the core particle becomes decorated with smaller nanoparticles. The importance of optimal concentration ratios for producing core–shell structures is also demonstrated with the synthesis of such structures of Ag/Si [77].

#### 5.1.3. Ag/TiO_2_ and Au/TiO_2_ Nanoparticles’ Formation Model

For the case of Ag and Au nanoparticles on TiO_2_, the differences in solubility of gold chloride and silver nitrate support the formation of the experimentally obtained final particle structures with this model. Following the proposed model, the precipitation of the gold chloride or silver nitrate starts when the critical supersaturation concentration is reached. As Ccrit_AuCl_3_ < Ccrit_AgNO_3_, the precipitation of gold chloride begins before silver nitrate at the given droplet diameter R″AuCl_3_ > R″AgNO_3_. The formation of the first metal clusters and the growth of noble metal occur on the droplet surface. As the oxide begins to form, the metal particles develop on the sites with high adsorption energy, which act as nucleation sites. After this, the precipitation line moves towards the centre of the droplet, and the diffusion of the precursor continues from the droplet centre towards the surface. As the precipitation of gold and silver follow the relation R″AuCl_3_ > R″AgNO_3_, the diffusion path for gold is longer and slower, until the radius reaches R‴ [37]. The result of the shorter path of silver is seen as less dispersion of this material across the oxide particle compared to gold, as silver has better conditions for reaching the droplet/particle surface before precipitating. This is also confirmed by the FIB milling analysis, showing less Ag nanoparticles in the TiO_2_ particle volume when compared to Au [36].

These results show that solubility is an important condition for phase separation when such particles are formed within the USP process. Figure 13 and Figure 14 show the morphological differences between the Ag/TiO_2_ and Au/TiO_2_ nanoparticles—the AuNPs are much finer than AgNPs, and are also distributed inside the volume of TiO_2_. The Ag nitrate precursor also has a melting point of 444 °C, lower than its temperature of decomposition, causing the AgNPs to grow on top of the oxide particles in melt form [37]. This also increases coagulation of these particles, resulting in larger AgNP size when compared to the AuNP sizes.

### 5.2. Au/Fe_2_O_3_ Nanoparticles

AuNPs have good potential for various applications, due to their properties such as Surface Plasmon Resonance and high biocompatibility. Adding ferromagnetism as an additional property to these nanoparticles enhances their applicability and use cases in medical treatment [78,79], for Magnetic Resonance Imaging [80], cancer treatment [81], drug delivery systems, as well as for catalysis, sensors, and so on [82,83,84,85]. Some initial experiments for determining the capability of producing such particles with USP were conducted, in order to investigate the formation mechanisms of these particles [86,87].

Different salt combinations containing Fe and Au were used for the USP starting solutions, in order to investigate the different formation mechanisms. The iron-containing salts were iron acetate, iron chloride and iron nitrate. Gold acetate, gold chloride and gold nitrate were used for the gold component. It is not possible to mix these salts in any given combination, as some react with one another in an aqueous solution, making this solution unusable in USP [86]. It was discovered that using different combinations of precursor salts produces particle shapes that correspond to the given salt. The iron chloride–gold chloride precursor was proven to be the most favourable for USP, producing spherical oxide particles, decorated with spherical and irregular metal particles. Other salt combinations produced irregular particles of the oxide, with irregular particles of the noble metal. Controlling the formation of these particles would be difficult, so only the chloride-chloride solution was pursued further for obtaining Fe oxide particles with uniformly dispersed AuNPs on their surface. The USP synthesis yielded mostly Au/Fe_2_O_3_ and Au/Fe_3_O_4_ nanoparticles, as the result of the initially chosen USP parameters. Using different parameters and temperatures for changing the iron oxide form into a singular Fe_3_O_4_ during synthesis achieved the magnetism characteristics of the particles, due to the different formation kinetics. This may be a subject for future endeavours with USP for the production of these particles.

In order to obtain nanoparticles with USP, the overall precursor concentration should be kept low, around 1–2 g/L [59,60,86]. Within this constraint, we have used iron chloride and gold chloride in different concentration ratios, ranging from Fe/Au = 0.1 to Fe/Au = 4, as seen in Figure 16.

A distinct particle morphology and AuNP arrangement difference is seen from the images of the prepared particles. When using a low concentration of iron in the precursor solution, the resulting particles have a greater number of larger AuNPs, which are more heavily agglomerated. As the ratio increases in favour of Fe, the AuNPs retain greater numbers, but become smaller and more uniform in shape [87].

A low ratio of Fe/Au 0.1 showed Fe oxide particles covered with larger clusters of AuNPs. When we increased the Fe/Au ratio to 0.25, lesser agglomeration of the AuNPs was visible, but the synthesised AuNPs were larger than when using the lower Fe/Au ratio. In the experiment Fe/Au 2, the AuNPs became finer and more evenly distributed on the Fe oxide particles, while still having a large number of irregular shapes and some agglomeration. With the Fe/Au ratio of four, the surfaces of Fe oxide particles were decorated with more uniformly dispersed finely sized AuNPs, with a measured mean nanoparticle size of 26 nm.

A comparison of the different particle morphologies obtained from the experiments is shown in Figure 17.

The observations from these experiments show that merely increasing Au content as compared to Fe content in the droplets does not produce more intrinsically uniform Au coatings, as was assumed previously. Following the previous metal/oxide formation model, the AuNPs are formed near the droplet surface, while the Fe oxide particles are formed inside the droplet core, due to the different solubilities and critical supersaturation conditions of the two components. As the droplet evaporates and shrinks, the surface chemistry of the system causes the AuNPs near the droplet surface to be deposited on the Fe oxide particles inside the core of the droplet. Some cases of Fe/Au core–shell synthesis report growth of Au on the Fe core, or growth of Fe on an Au shell, depending on the solvent type in which the growth took place, whether it was organic, water or otherwise [88,89,90]. The selection of solvent for the precursor preparation for USP synthesis may also be revised for continuous Au shell production [87].

For the Au/Fe_2_O_3_ system, the broad miscibility gaps between Fe and Au results in complete phase separation [89], while the adhesive forces between the Fe and AuNPs are much lower than the cohesive forces in the AuNPs. This results in the described Au/Fe_2_O_3_ particle morphologies observed in the investigation. The finding proposes that a continuous layer of Au on top of Fe oxide particles would not be possible to achieve with USP, without modifying the adhesive forces between the Fe oxide particles and Au [87].

There are also possibilities of using an intermediate layer between the metal and oxide, made with functional groups (citrates, thiols, amines, etc.), enabling the continuous Au shell growth [78,83,85,91].

## 6. Synthesis of Photoluminescent Nanoparticles

The need for phosphors, which absorb invisible light and emit visible light, becomes recognisable. Therefore, techniques including security inks or markers could be transported into wide applications, such as currency, food security, credit cards, passports, etc. [92,93]. Synthesis of appropriate inks, including a fluorescent component that could be activated with an ultraviolet or visible light, enables straightforward detection [94]. The type of phosphor materials is basically composed of a host matrix with intentionally doped impurities with enhanced light interaction [95,96]. Depending on the radiative emission mechanisms, photoluminescence can be categorised into two main groups; downconversion and upconversion, respectively. The former is the conversion of one high energy photon into two lower-energy photons, where the latter occurs through the conversion of multiple photons with lower energy into one high energy photon by Near-Infrared Light (NIR) excitation [97]. A host matrix should be thermally and chemically stable, optically transparent, and should not exhibit big size mismatch with a dopant ion. In order to fulfil these criteria, various materials have been examined by previous researchers as host matrices, such as: metal rare earth oxides, oxysulfides, fluorides, phosphates, vanadate [98]. Specifically, the excellent optical properties of Y_2_O_3,_ such as high refractive index (>1.9), broad transparency range (0.2–8 µm) and high light output, makes it an indispensable candidate for the luminescence applications [99]. For the surface plasmon enhancement, noble metal nanoparticles have been utilised, owing to their strong light absorption, originating from their surface plasmon’s interaction with the light. Surface plasmons are the oscillating waves propagating along the surface of the metal owing to electron clouds, and they produce an intense electromagnetic field. Coupling of the electromagnetic field created by plasmons and incident light can improve luminescence properties [100]. An alternative strategy is the surface plasmon enhancement of luminescence by incorporating metallic nanoparticles into phosphor materials. There have been many studies conducted on Eu complex solutions, including silver, and the effects of Ag nanoparticles on photoluminescent (PL) efficiency were investigated [101,102,103,104,105], even through USP [106]. In this research, it was shown that Ag/(Y_0.95_ Eu_0.05_ )_2_O_3_ nanocomposite can be synthesised by single USP at 800 °C temperature and 1.5 L/min air flow. For this purpose, Yttrium nitrate (Y(NO_3_)_3_.6H_2_O), europium nitrate (Eu(NO_3_)_3_∙5H_2_O) and silver nitrate (AgNO_3_) were used for preparation of the precursors. Deionised water was used as the solvent for all solutions. In USP synthesis, the precursor solutions were prepared by dissolving the relative amounts of Y(NO_3_)_3_.6H_2_O, Eu(NO_3_)_3_∙5H_2_O and AgNO_3_. The USP process starts with melting of the Y and Eu precursors, which provides a homogeneous mixture of both constituents. After that, they start to experience a stepwise thermal decomposition reaction simultaneously to form a hierarchical nanostructure. During their thermal decomposition, the Ag precursor starts to melt and, through the end of the Y and Eu nitrate decomposition reactions, the Ag nitrate starts to decompose. This explains the not well integrated and agglomerated Ag nanoparticles in the as-prepared samples (see Figure 18). Moreover, due to the fast heating and cooling rates, there were defects, pores and poor crystal arrangement in the as-prepared samples. During the next stage - heat treatment, enough time for diffusion was provided and better crystallisation and slight crystal growth were observed. Moreover, since Ag exists as a liquid at around 1000 °C, more homogeneous and smooth secondary particles (see Figure 19) were observed with respect to the as-prepared ones.

The mechanism given in Figure 20 was proposed considering the thermochemical and structural characterisation results. The USP process starts with melting of the Y and Eu precursors, which provides a homogeneous mixture of both constituents. After that, they start, simultaneously, to experience a stepwise thermal decomposition reaction to form a hierarchical nanostructure. During their thermal decomposition, the Ag precursor starts to melt and, through the end of the Y and Eu nitrate decomposition reactions, the Ag nitrate starts to decompose. This explains the not well integrated and agglomerated Ag nanoparticles in the as-prepared samples (See Figure 18). Moreover, due to the fast heating and cooling rates, there were defects, pores and poor crystal arrangement in the as-prepared samples. During the heat treatment, enough time for diffusion was provided and better crystallisation and slight crystal growth were observed. Moreover, since Ag exists as a liquid at around 1000 °C, more homogeneous and smooth secondary particles formed (see Figure 19). Moreover, a significant primary crystal growth of Eu^3+^:Y_2_O_3_ was observed, which implies that Ag did not hinder the mobility of grain boundaries, since it exists as liquid.

Consequently, due to the fast nature of the USP fast heating rates and shorter reaction times, heat treatment following synthesis is highly required to provide uniform distribution of the constituents. Higher Ag concentrations of the as-prepared samples resulted in emission quenching and yielded poor luminesce. However, Ag enhanced emission was observed within heat-treated samples. Then, 2.5 wt. % Ag incorporation followed by 2 h heat treatment at 1000 °C is reported as the most promising red light-emitting phosphors’ synthesis conditions via USP.

## 7. Conclusions and Outlook

From the presented review the following conclusions can be summarised:

(1) USP is a potentially reliable method for the large scale synthesis of nanoparticles; (2) The size of obtained nanoparticles is connected with the droplet diameter and the initial concentration of the precursor solution; (3) In USP, in general, there are two conversion formation mechanisms of nanoparticles: DTP and GTP; (4) Through USP it is possible to synthesise AuNPs, particles decorated with nanoparticles (Ag/TiO_2_ and Au/TiO_2_), Au/Fe_2_O_3_ nanoparticles, photoluminescent Ag/(Y_0.95_ Eu_0.05_)_2_O_3_ nanoparticles and others, in the form of powder or in solution as a suspension; (5) Characterisation of nanoparticles synthesised through USP is complex, and is connected closely to the different techniques, such as SEM, STEM, TEM, EDX analysis and others, with the aim of obtaining information about size, morphology, chemical composition and possible functional properties of nanoparticles.

It is important to take into account that the USP is a very fast process where all the described processes take place in micrometres’ volume and in parts of seconds. For these reasons, the deviations of the proposed models are possible, and, based on this, synthesised nanoparticles are very often in a metastable state. The speed of the process and its reaction mechanisms also prevent real-time observations of the particle formation, while the many chemical and physical species present in the reaction tube (simultaneously present droplets of various sizes, vapours of solvents and salts) make modelling of the formation mechanisms difficult. This requires multiple optimisation experiments in order to obtain the desired particle characteristics, as the material properties are affected by several parameters. The formation models are, as such, derived from the characterisation of the finally obtained particles, while the connections between the parameters and the material properties are frequently not complete. More advanced characterisation technologies could alleviate this issue and provide more insight into the formation mechanisms, in order to reduce the number of optimisation experiments. This is especially important with upscaled USP devices, for reducing the costs of production set up.

An advantage of USP is its ability to produce complex morphologies and compositions in a one-pot synthesis method, with a continuous production, and is a reasonable development direction of the USP. This ability gives USP the potential for greater uptake in electronic, energy devices, solar cells and photocatalysis [40,52]. However, complex particles again increase the number of required optimisation steps and experiments to obtain the desired particle properties.

The rapid formation at elevated temperatures may also result in undesired properties of the final particles, such as crystallinity, composition or morphology. In some cases, subsequent thermal or chemical treatment of the particles produces the desired result, however, this decreases the advantage of using USP as a one-pot synthesis method, increasing particle production complexity and costs. In some cases, optimisation of the USP parameters and formation conditions can produce more favourable results without additional processing.

## Figures and Tables

**Figure 1 materials-13-03485-f001:**
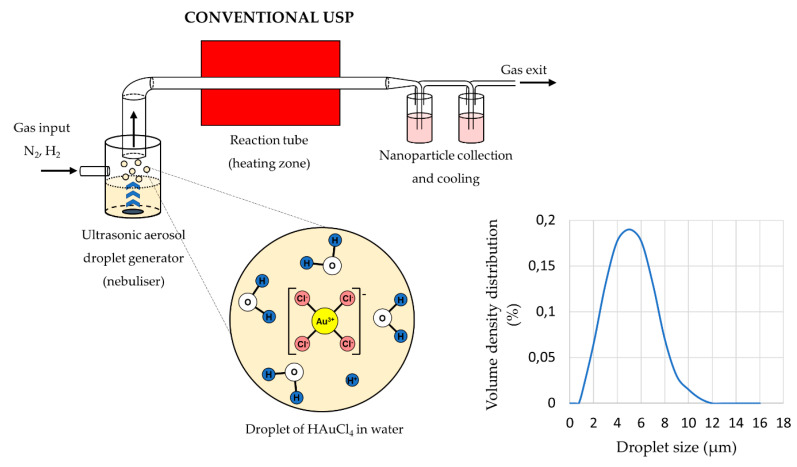
Aerosol droplet formation from a starting solution with ultrasound, an example of gold chloride precursor for AuNP synthesis (Assumption: The ion states inside the aerosol droplets are the same as in the starting solution, adapted from [46,47]).

**Figure 2 materials-13-03485-f002:**
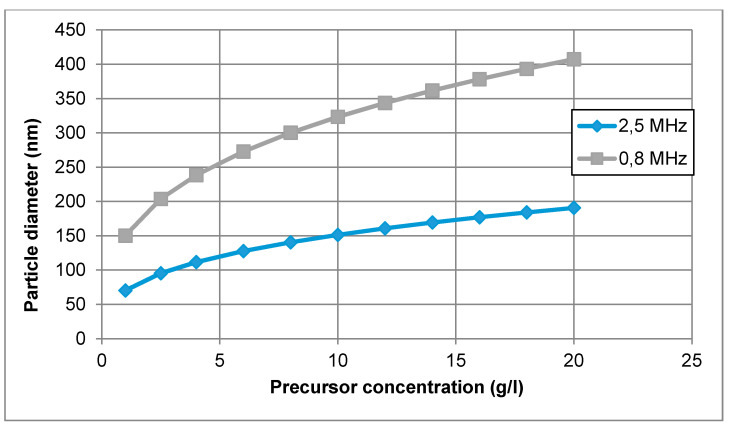
AuNP size calculations for increasing gold concentration in the precursor solution, for an ultrasound frequency of 0.8 and 2.5 MHz.

**Figure 3 materials-13-03485-f003:**
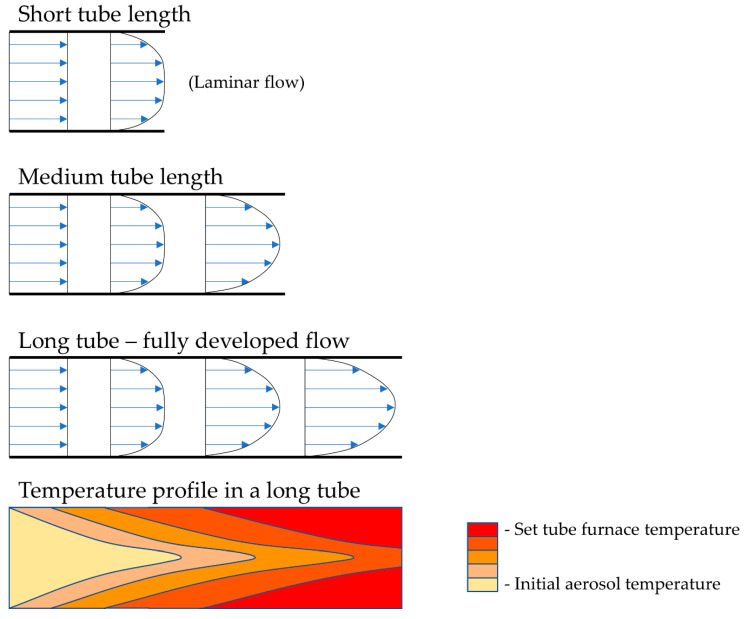
The gas velocity profile in the reactor tube for a laminar flow, depending on the tube diameter and length, with a representation of the temperature profile along the length of the tube.

**Figure 4 materials-13-03485-f004:**
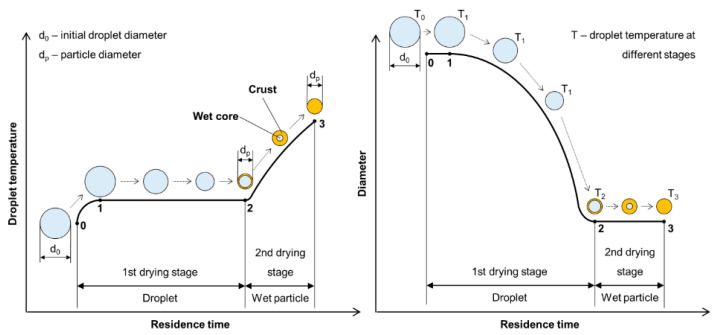
Evaporation of aerosol droplet and drying of the precipitated solute, adapted from [53].

**Figure 5 materials-13-03485-f005:**
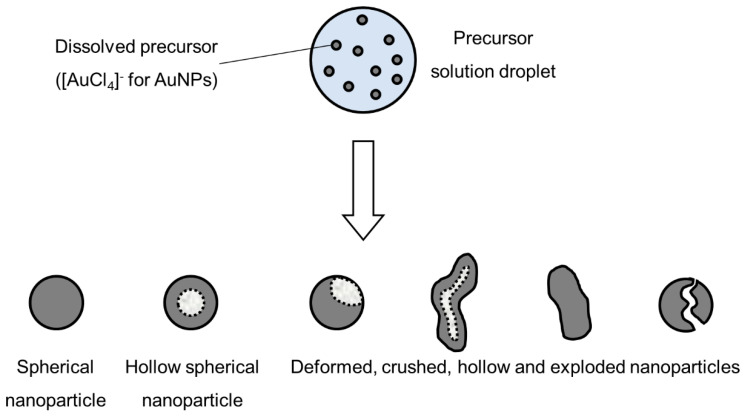
Types of nanoparticles that can be synthesised from an aerosol droplet, depending on the parameter conditions, adapted from [55].

**Figure 6 materials-13-03485-f006:**
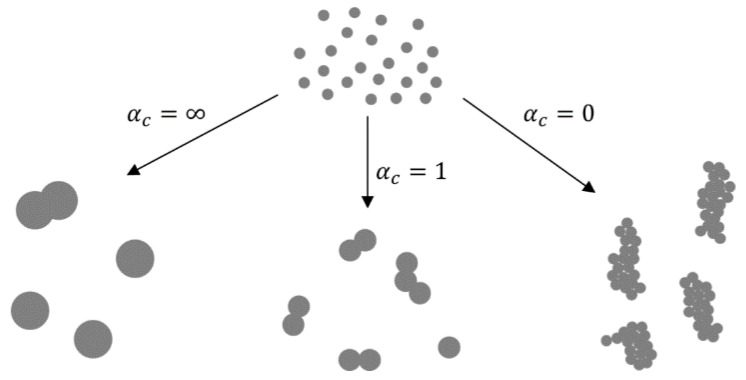
The ratios of collisions and coalescence with the Gas-To-Particle (GTP) mechanism and the resulting nanoparticle morphologies, adapted from [4,56].

**Figure 7 materials-13-03485-f007:**
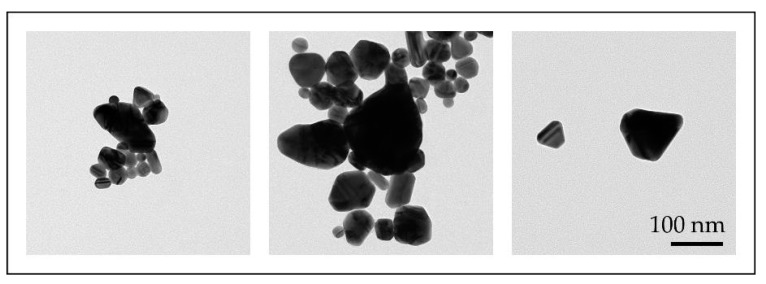
TEM images of AuNP shapes and sizes produced in a single batch with the conventional Ultrasonic Spray Pyrolysis (USP).

**Figure 8 materials-13-03485-f008:**
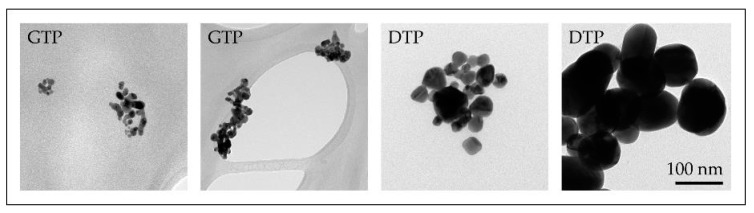
TEM images of AuNPs produced with the modified USP, the smaller nanoparticles (below and around 10 nm) were formed by the GTP conversion mechanism, while the larger AuNPs (below and above 100 nm) were formed by the Droplet-To-Particle (DTP) conversion mechanism.

**Figure 9 materials-13-03485-f009:**
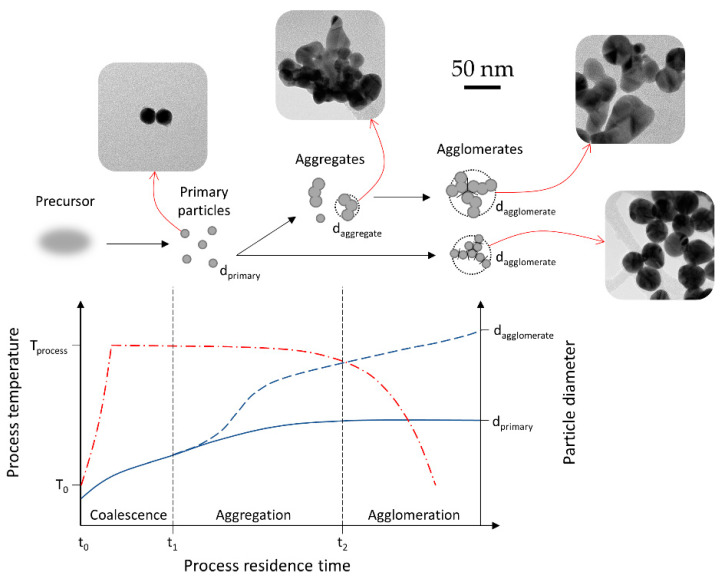
Overview of nanoparticle growth by coagulation and sintering depending on the process residence time, adapted for AuNP growth inside the USP process from the model presented in [10]. The red line represents the process temperature, while the blue line represents particle diameter.

**Figure 10 materials-13-03485-f010:**
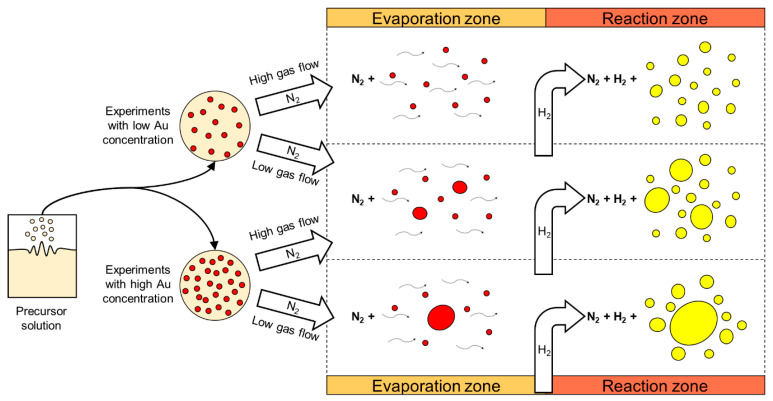
Practical formation model of AuNP synthesis in the modified USP. The typical morphologies are shown using high or low Au concentration in the precursor solution and high or low gas flow in the USP device. Adapted from [59].

**Figure 11 materials-13-03485-f011:**
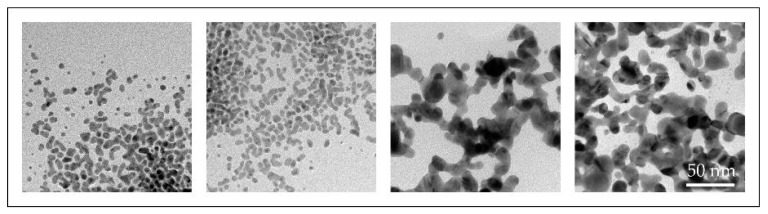
STEM images of AuNPs synthesised from Au acetate with the modified USP, showing the formation of a mesh-like structure composed of primary particles.

**Figure 12 materials-13-03485-f012:**
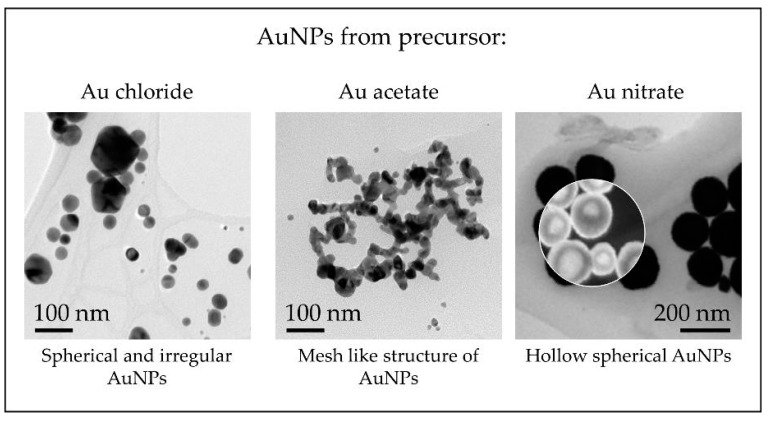
TEM overview of AuNPs synthesised with the modified USP showing the different nanoparticle structures obtained from the precursors used in the experimental investigations.

**Figure 13 materials-13-03485-f013:**
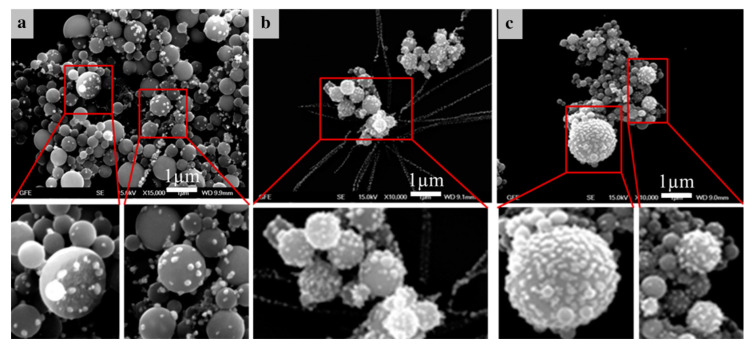
SEM images of the produced Ag/TiO_2_ nanoparticles, adapted from [36], (**a**) Ag content 20 wt.% with a reaction temperature of 800 °C, (**b**) Ag content 40 wt.% with a reaction temperature of 800 °C, (**c**) Ag content 40 wt.% with a reaction temperature of 1000 °C.

**Figure 14 materials-13-03485-f014:**
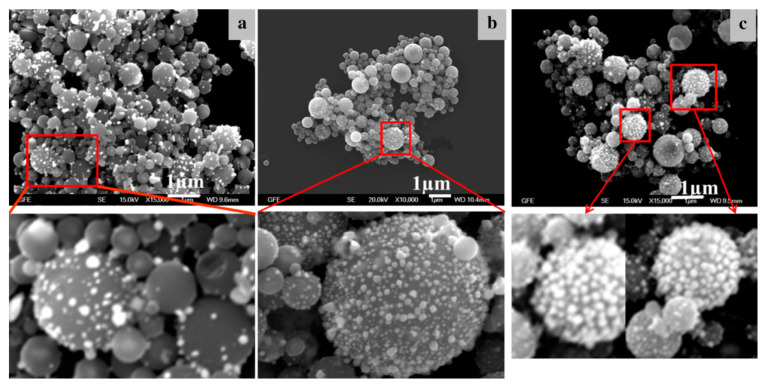
SEM images of the produced Au/TiO_2_ nanoparticles, adapted from [36], (**a**) reaction temperature of 550 °C, (**b**) reaction temperature of 800 °C, (**c**) reaction temperature of 1000 °C.

**Figure 15 materials-13-03485-f015:**
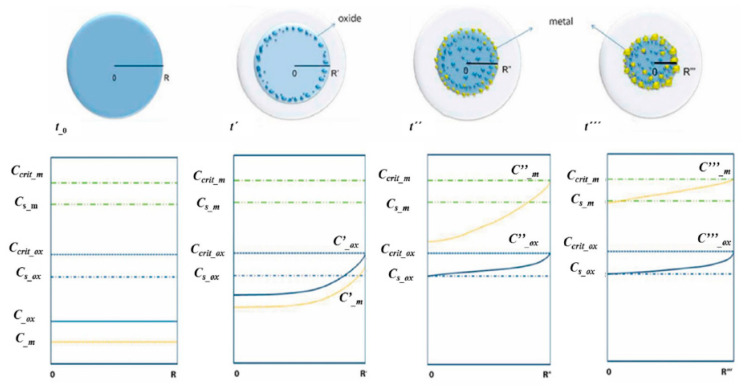
Metal/oxide nanoparticles’ formation model, adapted from [37].

**Figure 16 materials-13-03485-f016:**
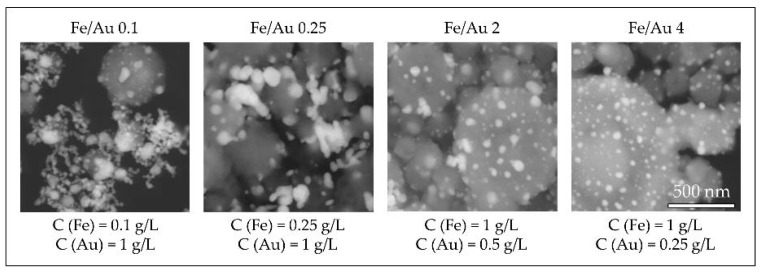
Comparison of the Au/Fe_2_O_3_ nanoparticles’ structures, synthesised with a precursor concentration ratio ranging from Fe/Au = 0.1 to Fe/Au = 4, adapted from [87]. The white particles represent AuNPs, the grey particles are Fe_2_O_3._

**Figure 17 materials-13-03485-f017:**
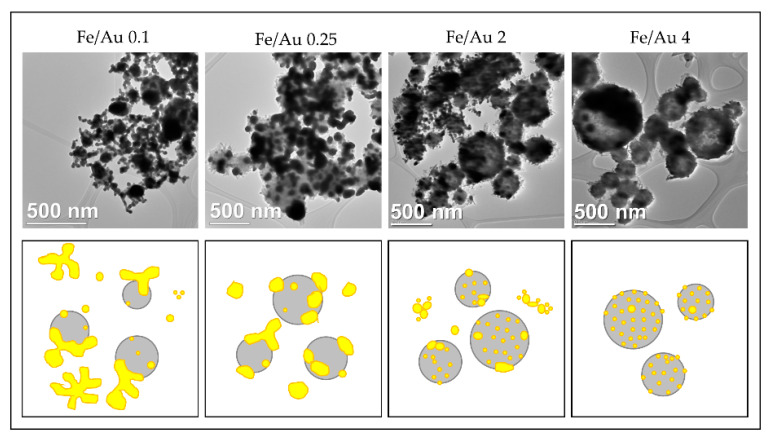
Schematic interpretation of the different Au/Fe_2_O_3_ nanoparticle morphologies obtained by using different Fe/Au concentration ratios with USP, adapted from [87].

**Figure 18 materials-13-03485-f018:**
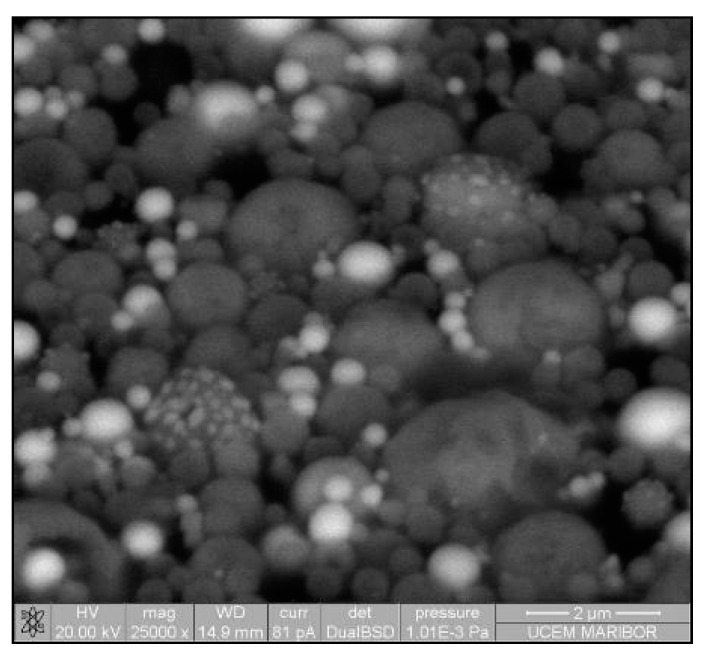
SEM image of USP synthesised Ag/(Y_0.95_ Eu_0.05_)_2_O_3_ nanocomposite.

**Figure 19 materials-13-03485-f019:**
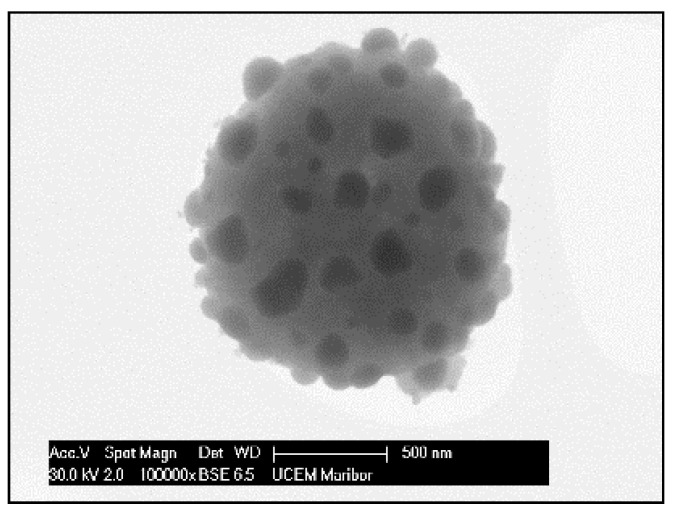
SEM image of heat-treated Ag/(Y_0.95_ Eu_0.05_)_2_O_3_ nanocomposite.

**Figure 20 materials-13-03485-f020:**
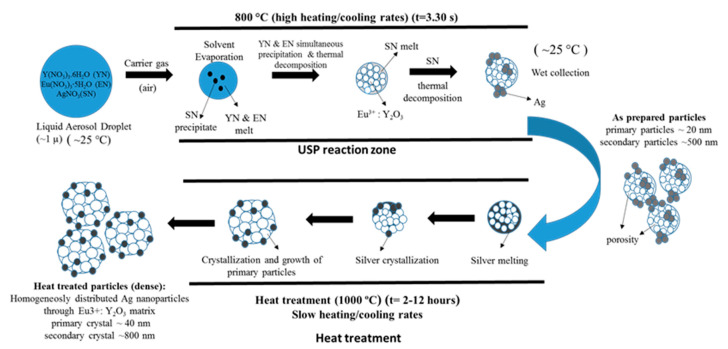
Formation mechanism of as-prepared and heat-treated Ag/(Y_0.95_ Eu_0.05_)_2_O_3_ nanocomposites adapted from [106].

**Table 1 materials-13-03485-t001:** Comparison of production yields of selected aerosol routes for particle synthesis.

Particle Production Method	Typical Particle Sizes	Possible Production Yield
Ultrasonic Spray Pyrolysis	Nanometer to micron-sized particles, 10 nm–5 µm	0.1–100 kg/d [28,39]
Flame Spray Pyrolysis	Nanometer to submicron-sized particles, 5–500 nm	1–25 kg/d, derived from [40]
Spray Pyrolysis—pneumatic nebuliser	Micron-sized particles or larger, >1 µm	>5 t/d [28]
Spray Pyrolysis—electrostatic nebuliser	Nanoparticles, <100 nm	<1 kg/d [28]

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
