# Peer review of "Advances in Ultrasonic Spray Pyrolysis Processing of Noble Metal Nanoparticles—Review"

_materials, 2020, doi:10.3390/ma13163485_

Round 1

Reviewer 1 Report

In this review, the authors summarize technological advances, trends and the scientific results of the Spray Pyrolysis in comparison with other similar processes. However, it does not discuss the current existing issues and the future development directions of spray pyrolysis processes for the fabrication of nanostructured. These contents should be added in the text. The text can be accepted by after major revision.

  1. The authors mentioned that the USP has advantage of large scale synthesis. However, there is no data such as production yield compared with other processes as various liquid or solid phase processes. The authors have to provide the comparison table for the production yield.
  2. The manuscript does not contain the latest scientific results and key findings for the synthesis of nanoparticles prepared by spray pyrolysis such as Chemical Engineering Journal368(2019)438-447, Nano-Micro Letters 11(2019)3. The authors should reference these related papers for advanced nanostructures and their processes.
  3. The authors should discuss the current existing issues and the future development directions of spray pyrolysis process. These contents should be added in the text.
  4. Some grammatical mistakes should be revised.

Author Response

  1. The authors mentioned that the USP has advantage of large scale synthesis. However, there is no data such as production yield compared with other processes as various liquid or solid phase processes. The authors have to provide the comparison table for the production yield.

We have added Table 1: Comparison of production yields of selected aerosol routes for particle synthesis, which compares related synthetic processes – pyrolysis.

  1. The manuscript does not contain the latest scientific results and key findings for the synthesis of nanoparticles prepared by spray pyrolysis such as Chemical Engineering Journal368(2019)438-447, Nano-Micro Letters 11(2019)3. The authors should reference these related papers for advanced nanostructures and their processes.

We have added some of the proposed references, along with others in the section describing the possibilities of producing different particle structures with USP.

  1. The authors should discuss the current existing issues and the future development directions of spray pyrolysis process. These contents should be added in the text.

Existing issues of the process have been added.

  1. Some grammatical mistakes should be revised.

The text of the article was reviewed again by the native speaker (17.7.2020 - Shelagh Baker). It is now free of any significant grammatical errors and is written in a clear and easily understandable style

Reviewer 2 Report

The manuscript entitled „Advances in Ultrasonic Spray Pyrolysis Processing of Materials – review” authored by Peter Majerič and Rebeka Rudolf is an extreme example of how review articles should not look like. I was interested in what authors have to say about the developments in the field, especially that I value research papers by Professor Rudolf. But the submitted manuscript is very disappointing, and there is no chance that it could be improved.

The review describes in any detail only research papers published by Peter Majerič and Rebeka Rudolf. There surely are some references to works by others. Still, they are usually just listed, e.g., as a possible application of particles of a specific type or other approaches (e.g., L583 “There are also possibilities of using an intermediate layer between the metal and oxide, made with functional groups (citrates, thiols, amines, etc.), enabling the continuous Au shell growth  [62,67,69,75].”). The merit of the review is based on 14 works of Professor Rudolf (8 among them were coauthored Peter Majerič) and not even single paper by other scientists. The Authors also use some phrases that make the manuscript looks like a part of a grant proposal, where they describe their accomplishments, e.g.:

L247 “In our case, gold or tetrachloroauric acid HAuCl4(s) was investigated the most extensively with USP synthesis”

L380 “As such, we have used USP to produce complex Ag/TiO2, Au/TiO2 and Au/Fe2O3 nanoparticles”

or

L86 “In our long‐term investigation and experimental work, we have used USP to produce several metallic and metal oxide particles with micron and nano sizes, ranging from Au (from Au salts and from gold scrap), Ag, Ag/TiO2, Au/TiO2, ZnO, Fe/Au, Co, Cr, and rare earth metals. For Au nanoparticles (AuNPs) we have also investigated extensively the cytotoxic and immunomodulatory properties of these particles for biomedical applications [23–25], as well as their behaviour when exposed to in situ heating during TEM observation [26], and their use in printing inks for Electronics (not yet published). We have also applied Au and ZnO nanoparticles in PMMA composites for advanced dental materials for removable prostheses [27]. The Ag/TiO2 and Au/TiO2 particles were produced for use as catalysts, and rare earth metals for photocatalytic applications [28,29].”

In my opinion, this is an outrageous attempt to narrow the field to the papers authored only by Peter Majerič and Rebeka Rudolf.

I do not see any other reason for the writing of such review other than 1) text was prepared for grant proposal and was reused as a manuscript and 2) making a publicity for other works by the Authors. The Authors did not even attempt to provide any comparison to other reviews in the field and to explain why the next review is needed (c.f. A comprehensive review on ultrasonic spray pyrolysis technique: Mechanism, main parameters and applications in condensed matter Journal of Analytical and Applied Pyrolysis Volume 141, August 2019, 104631). There is no attempt to give any future perspective and describe challenges that need to be overcome. There is very little analysis of the potential application and actual use of UPS in the industry (which is used in spite of what Authors wrote in the introduction).

Also, the claim from the abstract that “The present review shows some latest key findings in the field of USP” is false. After analysis of the list of references, it is clear that the review does not concern “latest key findings”. There are 0 references from 2020, only 6 from 2019 and 6 from 2018. More than 70% of all cited documents are 5 years old or more.

There are also some much smaller thing that I believe should be improved:

  • L103 “Ultrasonic nebulisers are the most efficient amongst other types of nebulisers, such as pneumatic and electrostatic.” Why? Describe in more details
  • The number of sentences is generic, e.g., “Nanoparticles of different materials have varied properties that make them interesting for implementation in novel products.” Please elaborate and give convincing examples.
  • No examples in 3.1 and 3.2 whatsoever is given.
  • Ambiguity in the number of statements, e.g., “depending on the process parameters used (reaction temperature, residence time, etc.).” What does this “etc.” means? Are there other important parameters? This problem occurs throughout the text.
  • In the “USP parameter selection” there is nothing about “gas flow (…) precursor solution salt and solvent volatility”, even despite it was promised earlier in the text. Also, there is very little on how “the tube diameter also dictates the velocity profile and temperature profile across the cross‐section of the tube”. I would expect to give newcomers some solid suggestions and not only to say that it is difficult to set the parameters and one needs to know how a change of one parameter influences others (L176 to 178). This is a lazy approach. Show some real advice.
  • There are two 5.1.2 sections.
  • There is zero references in the second 5.1.2 section.
  • 1.1 is broader than 5.1. In general, section 5 is disordered.
  • L263 “With initially smaller diameters of the aerosol droplets with diameters >1μm, nanoparticles are formed much sooner than with larger droplets with diameters <10μ” 5um satisfy both conditions. Not clear.
  • Fig 9 – what is a red curve? Temperature? The difference between aggregation and agglomeration is not adequately described.
  • 2, 6, 7, 8 – where they prepared for the paper? Were electron micrographs newer published before?
  • Fig 9 suggests that there are fractions from single precursor batch that have low or high concentrations. Was it intended?
  • Fig 2. – It would be nice to accompany these calculations with experimental results showing that the equation works.
  • In the numerous cases, AuNPs (I assume NP = nanoparticles) is used to describe objects of diameter larger than 100 nm. The most commonly used definition restricts nano to objects of at least a single geometrical dimension smaller than 100 nm.

There are also some problems with English.

  • Some paragraphs are only single sentences, e.g., L63 to L64 or L251 to L254. This is not how proper English text should be composed.
  • Some paragraphs are composed of just random thoughts, e.g.,:

“Currently there is a need for a scaled up production of nanoparticles, able to take up the increasing quantities of nanoparticles required for advanced applications. Technologies for generating nanoparticle powders and suspensions have existed for several decades [15] and are being improved upon continually, while novel approaches are also being studied [16,17]. One potentially reliable method for the large scale synthesis of nanoparticles is the USP process [18]. This process is considered to be relatively cost‐effective, and easily scalable from the laboratory to an industrial level. Several different types of nanoparticles and structures can also be produced, such as solid or hollow nanoparticles, core‐shell and ball‐in‐ball structures, etc. [4,19–22].”

The last sentence in this paragraph is entirely out of the blue here.

  • Spelling mistakes, e.g., L430 “The most stabile surface for…” L40 “Nanomaterials” (capital letters in the middle of the sentences) and many others.
  • L220 “a monomer (molecule or atom) is formed by a chemical reaction or a decrease in temperature” atoms are not created in conditions of USP.

Author Response

Please see the answers in the attached pdf. file!

Round 2

Reviewer 1 Report

The authors carefully revised the manuscript and the revised manuscript was updated to make the proposed concept more convincingly. In addition, English was polished carefully. Therefore, I would suggest accept of this manuscript.

Author Response

The authors carefully revised the manuscript and the revised manuscript was updated to make the proposed concept more convincingly. In addition, English was polished carefully. Therefore, I would suggest accept of this manuscript.

Thanks.

Reviewer 2 Report

I am not satisfied with the introduced changes. Two major issues were not fixed.

1) This is still a review of works by Rudolf et al. and not a comprehensive review of the topic. Literally all examples described in detail are originally authored by Professor Rudolf. This is not an "overview of the directions of research conducted with these materials" as claimed by Authors, but rather an overview of scientific interests in Rudolf lab.

2) Despite adding some new references (worth noting - one of two added works from 2020 is authored by Professor Rudolf) still, the review cannot be treated as up to date. Around 70% of all cited documents were published in 2015 or earlier (5y old or more). I did a quick count and I found references to 35 papers published in 2016, 2017, 2018, 2019, and 2020. This is less compared to the previously mentioned review publish over a year ago by Ardekani et al. who cited 40 papers published in 2016, 2017, 2018, and 2019 (obviously no 2020).

Author Response

1) This is still a review of works by Rudolf et al. and not a comprehensive review of the topic. Literally all examples described in detail are originally authored by Professor Rudolf. This is not an "overview of the directions of research conducted with these materials" as claimed by Authors, but rather an overview of scientific interests in Rudolf lab.

The reviewer is correct in the observation that the work does not cover a comprehensive review of spray pyrolysis. The work rather focuses on noble metals (namely Ag and Au), and noble metal composite particle production. This inacurracy was changed to reflect this focus.

2) Despite adding some new references (worth noting - one of two added works from 2020 is authored by Professor Rudolf) still, the review cannot be treated as up to date. Around 70% of all cited documents were published in 2015 or earlier (5y old or more). I did a quick count and I found references to 35 papers published in 2016, 2017, 2018, 2019, and 2020. This is less compared to the previously mentioned review publish over a year ago by Ardekani et al. who cited 40 papers published in 2016, 2017, 2018, and 2019 (obviously no 2020).

Thank you for carefully reviewing the corrections and additions. Older references were included in the work, as they were considered applicable in order to remain inside the intended focus and point of view. We leave the decision to publish the review to the editor.

The following references were added in the revision, which were not adequately highlighted previously:

  1. Oh, S.H.; Jo, M.S.; Jeong, S.M.; Kang, Y.C.; Cho, J.S. Hierarchical yolk-shell CNT-(NiCo)O/C microspheres prepared by one-pot spray pyrolysis as anodes in lithium-ion batteries. Chemical Engineering Journal 2019, 368, 438–447, doi:10.1016/j.cej.2019.02.144.
  2. Jo, M.S.; Ghosh, S.; Jeong, S.M.; Kang, Y.C.; Cho, J.S. Coral-Like Yolk–Shell-Structured Nickel Oxide/Carbon Composite Microspheres for High-Performance Li-Ion Storage Anodes. Nano-Micro Lett. 2019, 11, 3, doi:10.1007/s40820-018-0234-0.
  3. Oh, S.H.; Kim, J.K.; Kang, Y.C.; Cho, J.S. Three-dimensionally ordered mesoporous multicomponent (Ni, Mo) metal oxide/N-doped carbon composite with superior Li-ion storage performance. Nanoscale 2018, 10, 18734–18741, doi:10.1039/C8NR06727A.
  4. Cho, J.S.; Ju, H.S.; Lee, J.-K.; Kang, Y.C. Carbon/two-dimensional MoTe2 core/shell-structured microspheres as an anode material for Na-ion batteries. Nanoscale 2017, 9, 1942–1950, doi:10.1039/C6NR07158A.
  5. Cho, J.S.; Lee, S.Y.; Lee, J.-K.; Kang, Y.C. Iron Telluride-Decorated Reduced Graphene Oxide Hybrid Microspheres as Anode Materials with Improved Na-Ion Storage Properties. ACS Appl. Mater. Interfaces 2016, 8, 21343–21349, doi:10.1021/acsami.6b05758.
  6. Leng, J.; Wang, Z.; Wang, J.; Wu, H.-H.; Yan, G.; Li, X.; Guo, H.; Liu, Y.; Zhang, Q.; Guo, Z. Advances in nanostructures fabricated via spray pyrolysis and their applications in energy storage and conversion. Chem. Soc. Rev. 2019, 48, 3015–3072, doi:10.1039/C8CS00904J.
  7. L, R.T.H.; Escalona, A.; S, J.G.; Sánchez, A.; Orozco, S. Ultrasonic spray pyrolysis synthesis of Al, Zr, and Ti oxides multishell. J. Phys.: Conf. Ser. 2019, 1221, 012026, doi:10.1088/1742-6596/1221/1/012026.
  8. Tran, M.N.; Cleveland, I.J.; Aydil, E.S. Resolving the discrepancies in the reported optical absorption of low-dimensional non-toxic perovskites, Cs3Bi2Br9 and Cs3BiBr6. J. Mater. Chem. C 2020, doi:10.1039/D0TC02783A.
  9. Manivasakan, P.; Karthik, A.; Rajendran, V. Mass production of Al2O3 and ZrO2 nanoparticles by hot-air spray pyrolysis. Powder Technology 2013, 234, 84–90, doi:10.1016/j.powtec.2012.08.028.
  10. Mueller, R.; Mädler, L.; Pratsinis, S.E. Nanoparticle synthesis at high production rates by flame spray pyrolysis. Chemical Engineering Science 2003, 58, 1969–1976, doi:10.1016/S0009-2509(03)00022-8.
  11. Jung, D.S.; Park, S.B.; Kang, Y.C. Design of particles by spray pyrolysis and recent progress in its application. Korean J. Chem. Eng. 2010, 27, 1621–1645, doi:10.1007/s11814-010-0402-5.
  12. Rahemi Ardekani, S.; Sabour Rouh Aghdam, A.; Nazari, M.; Bayat, A.; Yazdani, E.; Saievar-Iranizad, E. A comprehensive review on ultrasonic spray pyrolysis technique: Mechanism, main parameters and applications in condensed matter. Journal of Analytical and Applied Pyrolysis 2019, 141, 104631, doi:10.1016/j.jaap.2019.104631.
  13. Tiyyagura, H.R.; Majerič, P.; Anžel, I.; Rudolf, R. Low-cost synthesis of AuNPs through ultrasonic spray pyrolysis. Mater. Res. Express 2020, 7, 055017, doi:10.1088/2053-1591/ab80ea.
  14. Sirelkhatim, A.; Mahmud, S.; Seeni, A.; Kaus, N.H.M.; Ann, L.C.; Bakhori, S.K.M.; Hasan, H.; Mohamad, D. Review on Zinc Oxide Nanoparticles: Antibacterial Activity and Toxicity Mechanism. Nanomicro Lett 2015, 7, 219–242, doi:10.1007/s40820-015-0040-x.
  15. Jiang, Z.-J.; Liu, C.-Y.; Sun, L.-W. Catalytic Properties of Silver Nanoparticles Supported on Silica Spheres. J. Phys. Chem. B 2005, 109, 1730–1735, doi:10.1021/jp046032g.
  16. Jung, D.S.; Ko, Y.N.; Kang, Y.C.; Park, S.B. Recent progress in electrode materials produced by spray pyrolysis for next-generation lithium ion batteries. Advanced Powder Technology 2014, 25, 18–31, doi:10.1016/j.apt.2014.01.012.

Round 3

Reviewer 2 Report

I understand that the Editor is in favor of this review. The issue of the described research articles (all examples published by Rudolf et al.) is not viewed by the Editor as a problem. However, in my humble opinion, this should not be published.

Author Response

Dear Editor,

Please find enclosed with this submission the improved manuscript titled “Advances in Ultrasonic Spray Pyrolysis Processing of Noble Metal Nanoparticles – review” as suggested – new tittle and new Abstract. All new improved text is in red.

Note:

The proofreader responsible for the English language is Shelagh Hedges, Faculty of Mechanical Engineering, University of Maribor, Slovenia.

Best Regards,

Yours faithfully,

Rebeka Rudolf

On behalf of the authors.
